# Correction of Threshold Determination in Rapid-Guessing Behaviour Detection

Muhammad Alfian [1], Umi Laili Yuhana [1], Eric Pardede [2],* and Akbar Noto Ponco Bimantoro [1]

1 Institut Teknologi Sepuluh Nopember, Surabaya 60111, Indonesia; ini.muhalfian@gmail.com (M.A.); yuhana@if.its.ac.id (U.L.Y.); akbarnotopb@gmail.com (A.N.P.B.)
2 Department of Computer Science and Information Technology, La Trobe University, Melbourne, VIC 3000, Australia
* Correspondence: e.pardede@latrobe.edu.au

**Abstract:** Assessment is one benchmark in measuring students' abilities. However, assessment results cannot necessarily be trusted, because students sometimes cheat or even guess in answering the questions. Therefore, to obtain valid results, it is necessary to separate valid and invalid answers by considering rapid-guessing behaviour. We conducted a test to record exam log data from undergraduate and postgraduate students to model rapid-guessing behaviour by determining the threshold response time. Rapid-guessing behaviour detection is inspired by the common k-second method. However, the method flattens the application of the threshold, thus allowing misclassification. The modified method considers item difficulty in determining the threshold. The evaluation results show that the system can identify students' rapid-guessing behaviour with a success rate of 71%, which is superior to the previous method. We also analysed various aggregation techniques of response time and compared them to see the effect of selecting the aggregation technique.

**Keywords:** rapid-guessing behaviour; threshold determination; response time

## 1. Introduction

In fact, assessment plays a very important role in the learning process [1]. Assessment is a process of evaluating knowledge, the ability to understand, and achievement of test takers' skills [2]. Assessment is used to measure students' abilities with the aim of selecting students for new admissions, measuring the level of understanding of post-learning material, and as a determinant of graduation. In addition, one of the benefits of conducting an assessment is as a reference for determining student learning flows. An example is the determination of material according to students' abilities [3] and determining the next material they need to study [4]. In addition, student assessments can streamline the allocation of resources needed to increase student learning competencies [5].

As test-takers, we often do not know whether these students' answers are valid or not, and whether they are taking it seriously or cheating. As students, we also sometimes come across questions that are very difficult, forcing us to answer to obtain the best grades even though we do not know the answers. This behaviour is called rapid-guessing behaviour. According to ref. [6], rapid-guessing behaviour occurs when test takers answer questions quicker than usual in a speeded test. However, assessment results can be invalid because students cheated or rapidly guessed the answer to the question [6]. Ref. [7] states that, therefore, to obtain the ideal assessment results, it is necessary to differentiate assessment results based on student behaviour, whether they answer by guessing (rapid-guessing behaviour) or answer seriously (solution behaviour). This rapid-guessing behaviour causes biased scores and unreliable tests, so it should be ignored.

Schnipke was the first to discover rapid-guessing behaviour when mapping the response times of the Graduate Record Examination Computer-Based Test (GRE-CBT). In her research, each question was mapped to its response time distribution as shown in

Figure 1. Response time is taken from how long it takes students to read to answer a question. In practice, to distinguish rapid-guessing behaviour and solution behaviour, we need to determine the threshold time.

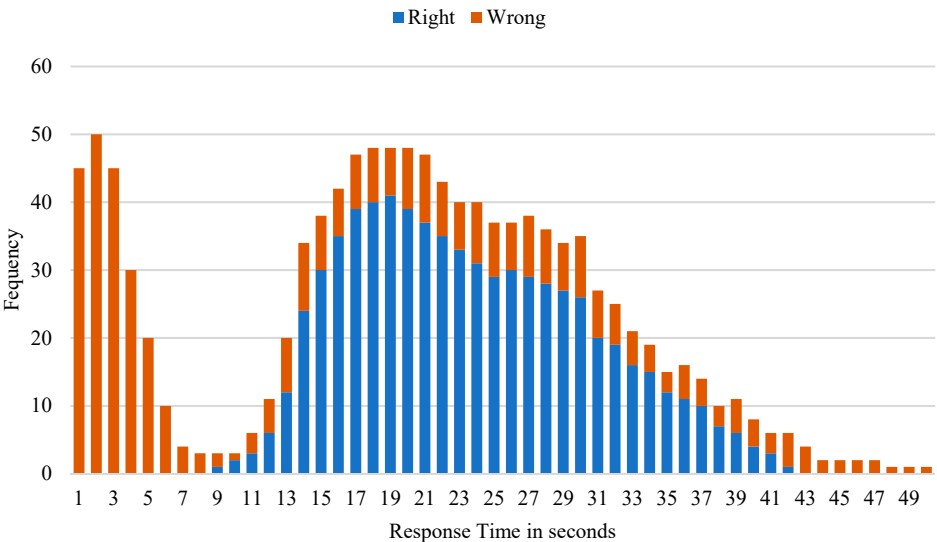

**Figure 1.** Example of RT distribution.

Several studies investigated how to determine the threshold time. Schnipke use visual inspection to determine threshold and distinguish both behaviours. A similar approach was carried out by DeMars [8], Setzer et al. [9], and Pastor et al. [10]. However, detecting rapid-guessing behaviour becomes more difficult using this approach when the RT distribution has the same RT peak. Students who answered by guessing and students who answered seriously both made overlapping response time distributions. Other researchers used the k-second method to determine the RT threshold, in which the fixed threshold value is generally set between three to five seconds [7]. K-second is the simplest threshold method. It does not require information about each item's surface features or response time distribution and is particularly useful with large item pools. Its one-size-fits-all nature, however, will often result in variations in misclassification across items [7–10].

Some other researchers use the surface features method to distinguish between the two behaviours. Surface features determine the RT threshold using several item features. Silm et al. [11] considered the test subject and item length in determining the RT threshold. Wise and Kong [12] considered the number of characters and whether there were tables or images. However, in both studies, the results of evaluating students' rapid guessing behaviour were not explicitly detailed. In contrast to methods that use time thresholds, Lin [13] processes the student's ability score (l) and item difficulty index (i) based on the Rasch model to determine guessing behaviour. They argue that if there is a large difference between the student's ability and the item difficulty index, then it is rapid-guessing behaviour.

This study aims to propose a correction to the determination of time thresholds as part of the identification of rapid-guessing behaviour in assessment. The correction we provide is that the determination of the threshold is not simply about how to choose the right number to be used as a threshold, but also needs to pay attention to how difficult the question and how the data processing technique is. We tried several data aggregation techniques such as sum, average, and maximum. We adopted the concept of k-seconds and combined it with the features of item response theory (IRT) to create a new approach in determining the time threshold for each item category. The questions were divided into three categories based on their difficulty according to IRT features. Data was obtained from online exams during lectures on campus. Response time is obtained from how long students work on questions (calculated from the time of opening to answering questions).

The expected benefit of this research is that the question maker can know which answers are given seriously by students and which are given fraudulently, so that the scores can be differentiated. This research is part of our larger research on computer adaptive assessment.

## 2. Related Works

Rapid-guessing behaviour is a phenomenon when students answer items rapidly without serious thought. In other words, students randomly guess the answers to the items. Rapid-guessing behaviour usually occurs in multiple choice tests. We discussed how rapid-guessing behaviour is detected in exams. There have been several variables used to detect rapid-guessing behaviour. The most popular approach is rapid-guessing detection based on response time (RT). Other variables include student ability, item difficulty, and response accuracy (RA). In the next section we discuss our proposed method and our contribution to rapid-guessing behaviour detection.

### 2.1. Detection Based on Response Time (RT)

Schnipke [6] is one of the first researchers that used RT thresholds as the basis for detecting rapid-guessing behaviour. Visual inspection was carried out on RT distributions of 17,415 students that took a computer-based Graduate Record Examinations Computer-based Test (GRE-CBT). The RT of correct and wrong responses for each item were separately plotted to visualize the distribution of RT of each item. In this study, rapid-guessing behaviour towards an item is indicated by a larger number of fast wrong responses in the RT distribution of the item. Figure 1 shows the distribution of two items, in which wrong responses are indicated by the red lines. In the first distribution, the RT for majority of the students is relatively short, and the number of wrong responses exceed the number of correct responses. While in the second distribution, the RT for majority of the students is relatively long and the number of correct responses exceeds the number of wrong responses. Therefore, the first distribution is classified as rapid-guessing behaviour and the second distribution is classified as solution behaviour (students fully consider the answer). Furthermore, in the second distribution that is classified as solution behaviour, the fastest RT of a correct response is five seconds; therefore, a RT under five seconds is rapid-guessing behaviour.

A similar approach was carried out by DeMars [8], Setzer et al. [9], and Pastor et al. [10]. However, detecting rapid-guessing behaviour becomes more difficult using this approach when RT distributions classified as rapid-guessing behaviour and solution behaviour possess similar peaks of RT. This is because the time needed to correctly answer items is indeed short.

Other researchers used the k-seconds method to determine the RT threshold, in which the fixed threshold value is generally set between three to five seconds [7]. The threshold value was then used to determine the response time effort (RTE) of the students. Wise [7] evaluated the proposed RTE model on students that were given mathematics and reading tests in varying times, days, seasons, and age groups. From the experimental results, it was indicated that RTE is influenced by several factors, namely gender, age, contents of an item, and time.

### 2.2. Detection Based on Combination of RT and Other Variables

Surface features is a method used to determine the RT threshold using several item features. Unlike the k-seconds method that sets the same RT threshold value to all the items, in the surface features method, each item is given an RT threshold based on its features. The features include the number of characters in an item, whether an item consists of tables and figures, and the subject being evaluated by the item. Several features that were used in previous studies and the resulting RT threshold values are shown in Table 1. Silm et al. [11] considered the subject of the test and item length in determining the RT threshold as shown in Table 1. Wise and Kong [12] took into consideration the number of characters and whether an item consisted of a table or figure in the determination of the RT

threshold. However, in both studies, the results of rapid-guessing behaviour evaluation on the students were not explicitly detailed.

**Table 1.** Surface Feature Threshold.

| Criteria | Threshold |
|---|---|
| Math/spatial reasoning problem | 5 s |
| <200 characters | 3 s |
| 200–1000 characters | 5 s |
| >1000 characters | 10 s |

Pastor et al. [10] used latent class analysis (LCA) to investigate whether there was a difference in solution behaviour patterns across three tests differing in content. They implemented the RT threshold value resulting from visual inspection of RT distributions into the LCA model. From the experiment that was carried out on undergraduate students, it was found that the results of the proposed method were similar to that of Wise et. al. [14], in which the solution behaviour pattern is consistent in all the tests differing in content. The experimental results were validated using the BCH approach (Bolck, Croon, and Hagenaars [15]), which involves performing a weighted ANOVA, with weights that are inversely related to the classification error probabilities [16].

Another study, proposed by Lee and Jia [17] combined RT and RA to determine the time threshold. Time thresholds were determined based on the participants' RTs for test 1 and test 2, as shown in Figure 2. The RT results of each test were then combined to be analysed manually using either common k-seconds or visual inspection of the RT distribution. The test was conducted on approximately 8400 junior high school students in mathematics with a composition of 40% students in a multistage test (MST) sample and 60% students in control sample. The proposed method is evaluated manually by the authors with expert inspection of the questions, such as the presence of tables or figures and the complexity of the questions.

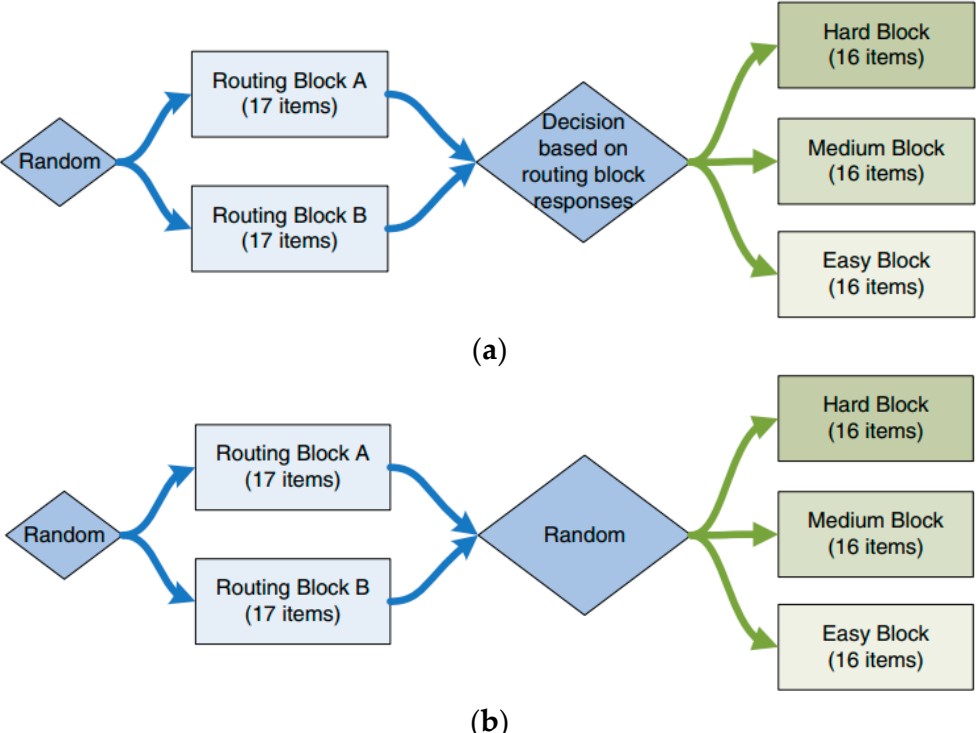

**Figure 2.** Test scenario using multistage test (MST) (**a**) and control test (**b**).

In contrast to the method that uses a time threshold, Lin [13] processed the value of the student's ability ($l$) and the difficulty index ($i$) based on the Rasch model to determine guessing behaviour. Student's ability ($l$) refers to the measure of how proficient a student is, and difficulty index ($i$) refers to the measure of how hard an item (question) is. They argue that if there is a big difference between the logit ability and the difficulty index of the question, it should be rapid-guessing behaviour. They classified the answers as rapid-guessing behaviour if $l - i \leq 2$. Answers that were classified as rapid-guessing behavior were removed from the dataset and used as the final test model on the language test of sixth-grade elementary school students. From the tests carried out, they found that the assessment of high-ability students had better precision.

Based on previous literature studies, no research has developed and corrected time threshold determination utilising IRT features and considering variations in data aggregation. Therefore, this study aims to combine the k-second method with IRT features to recognize the difficulty level of each question and utilise multiple data aggregation methods to distinguish rapid-guessing behaviour and solution behaviour. We compared the proposed method with previous methods such as the common k-second, surface, and normative. We pay attention to the data aggregation technique, because in the classification process it is not only about how to determine the right threshold value, but also the aggregation technique is also important. Some of the aggregation techniques we used include average, sum, and maximum. Then, the model is evaluated using accuracy, precision, recall, and F1 score parameters. The next section will describe this method in more detail.

## 3. Methods

This section details the methodology used for detecting rapid-guessing behaviour. As shown in Figure 3, this research consists of two main processes: a conventional test and rapid-guessing modelling.

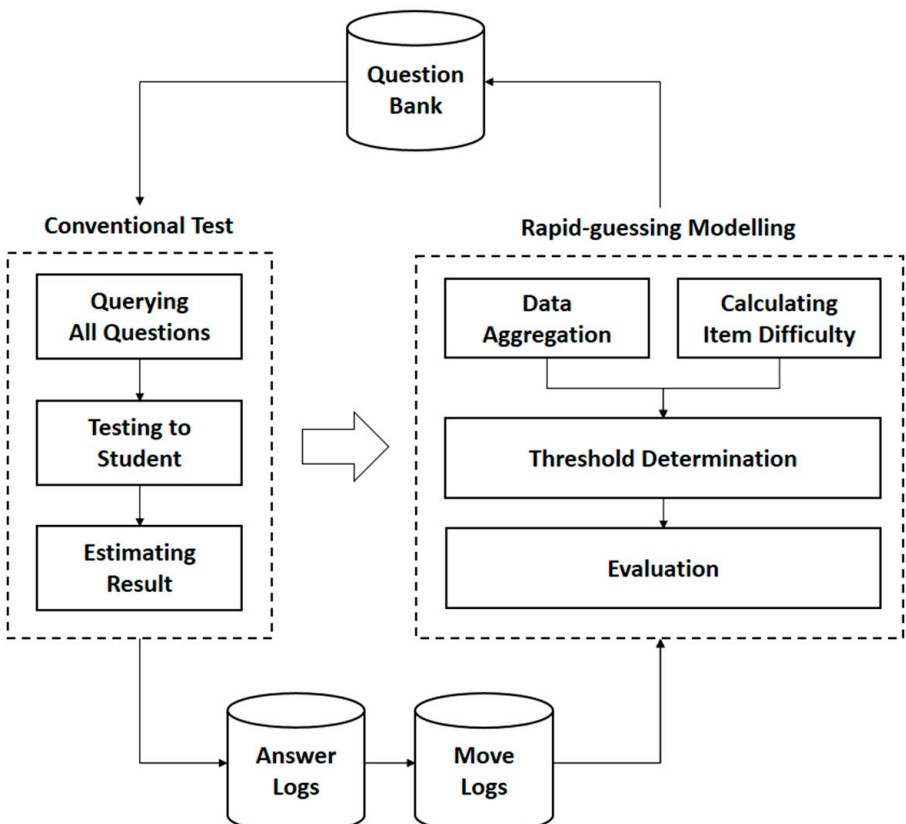

**Figure 3.** System design for detecting rapid-guessing behaviour.

### 3.1. Gathering Data from Conventional Test

One of the advantages of computer-based tests (CBTs) is that data on the student's activities from the start to the end of the test can be easily obtained. The data is accessible, provides meaningful information, and is unambiguous because every student has their own accounts and all activities of students are recorded. This study focuses on analysing user behaviour data from the system log, without taking into consideration demographic factors such as age, gender, and ethnicity of the students which may cause the proposed model to become biased towards these factors.

This study analyses student daily test data in a specific course. The examinees of the daily test are university students that are technologically literate. The students were first given a conventional test. The conventional test consisted of 40 items. The questions were multiple choice with one correct answer. All students worked on the same questions at the same time. This was so as to evaluate the comprehensive ability of the students in understanding the study material. Furthermore, the results of the comprehensive test are used to calculate item difficulty of each item in the test.

The platform used for this test is a web-based "i-assessment" software accessed through smartphones. The "i-assessment" software records student activity during the test and the answers of the students and stores the data in a database. The time a student accesses a question and the time the student answers the question is stored in the Answer Log table. Furthermore, the time a student navigates between questions is stored in the Move Log table. Every time a student gives an answer to each question, a pop up appears in the system asking, "Are you sure about your answer?". We use this data as a reference to distinguish answers that are guessing and not.

### 3.2. Conventional Test Information

The tests were administered to students of a widely recognized university in Indonesia. The students were given an end-of-semester daily test (quiz) by the lecturer. The detailed information is shown in Table 2. The test data was collected from two courses, namely software project management (SPM) and software engineering (SE). The SPM course is an undergraduate course, while the SE course is a postgraduate course. The duration of the conventional test was 90 min and consisted of 40 multiple-choice items, in which each item presented five answers to choose from. The average scores of each course showed that students in the SPM course had a fairly high score, as seen from the average score of 65.89. In contrast to students in SE courses, students have fewer high scores, as seen from the average score of 44.58. Even though the standard deviation of the SPM test scores was higher than that of the SE test scores, the minimum and maximum score were higher for the SPM test. However, these data alone are insufficient to adequately assess the educational evaluation process. Further analysis needs to be carried out with respect to the test items and other underlying factors of the students.

**Table 2.** Data summary.

| Course and Duration | Class Member | Level | Score |
|---|---|---|---|
| SPM 90 min | 45 students | Undergraduate | Mean = 65,89 Std = 15 Min = 38 Max = 93 |
| SE 90 min | 45 students | Graduate | Mean = 44,58 Std = 11,82 Min = 27,5 Max = 75 |

### 3.3. Rapid-Guessing Modelling

The first step in rapid-guessing modelling is data aggregation. This stage combines data from several tables into a single unit. Both the Answer Log table and the Move Log

table possess a relationship with the Participant table and the Question table. The Answer Log table stores information on when students open a question, and when they answer the question. Meanwhile, the Move Log table stores information on when students moved from one question to another, regardless of when they answered the question. After gathering the relevant data, the Log Aggregation table is generated to store a summary of data of both the Answer Log and Move Log tables based on the key attributes of the log tables. This transformation process is called data aggregation. Data aggregation is the process of finding and gathering data and visualizing the data in a summarized format for an easier statistical analysis of the data. The Log Aggregation table possesses columns that are produced from the aggregation process, including sum, maximum, minimum, and average values as shown in Figure 4. The Log Aggregation table is then split with respect to the purpose of the data analysis based on questions, participants, and a combination of both.

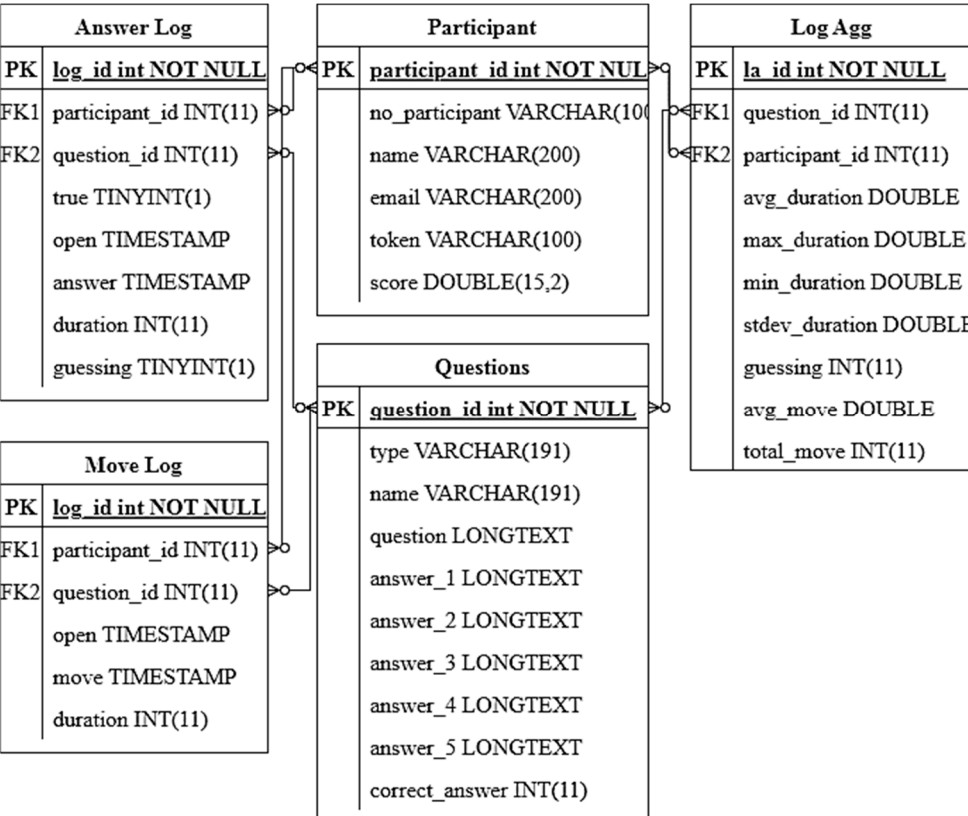

**Figure 4.** Aggregation table diagram.

The second step is calculating item difficulty. Item difficulty ($b_i$) is defined as the proportion of examinees that were able to correctly answer the item [18]. Item difficulty in item response theory(IRT) is derived from the z-score measurement method. Therefore, item difficulty is calculated by dividing the number of examinees that were unable deliver correct answer to item i ($n_{fi}$) by the total number of examinees that submitted a response item i ($N_i$) minus the number of examinees that were unable to submit (false answer) a response to item i ($n_{fi}$). The resulting value is then normalized using the natural logarithm to decrease the distribution value [19], as shown in Equation (1).

$$b_i = \ln\left(\frac{n_{fi}}{N_i - n_{fi}}\right) \tag{1}$$

After that, from the question difficulty values, we categorised the questions into three labels, namely easy, medium, and difficult, based on the question difficulty parameters in

IRT. We labelled them using the fuzzy logic inference method. Figure 5 shows the member function of item difficulty. The y-axis shows the fuzzy inference value, while the x-axis value shows the item difficulty value. The range of item difficulty values is from $-3$ to 3. For this question, we directly divided it into three labels. The easy label is given if the item difficulty ranges from $-3$ to 0. Meanwhile, the medium label is given if the item difficulty ranges from $-1$ to 2. And finally, the difficult label is given if the item difficulty level is above 1. These three different labels are to categorise student responses, and then determine the threshold for each item label.

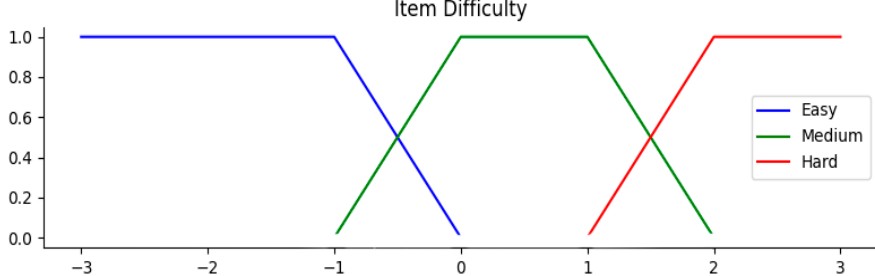

**Figure 5.** Item difficulty member function.

The third step is determining the threshold. We are inspired by the common k-seconds [7] method to determine the threshold. While the common k-seconds method sets all questions with the same threshold, we have a different approach. We categorize items into three labels based on their difficulty level, namely easy, medium, and hard. Each question label has its own threshold. The determination of the threshold is the same as the predecessor method, which is that we use a common value and then match with the dataset which value is the best. The value we agreed on was 3 s for questions with the hard label, and 2 s for questions with the easy and medium labels.

The last step is evaluation. We evaluate the model by calculating the evaluation matrix. We compare our proposed method with previous methods. In addition, we also compare various aggregation techniques, so that we can find out the effect of different aggregation techniques on the classification results.

## 4. Results and Discussion

We conducted experiments on students during lecture hours. Some of the steps were aggregating data, calculating item difficulty, determining threshold, and evaluation. The first step is to perform data aggregation. We collect data from the Answer Log and Move Log tables to be aggregated in an aggregation table according to the design. However, the parameters we use here are only time-related parameters, including avg_duration, max_duration, min_duration, stdev_duration, avg_move, and total_move. However, considering the processing time, we chose three main parameters to compare, namely avg_duration, max_duration, and total_move.

The second step is to calculate item_difficulty. We use the equation from IRT to calculate the item difficulty. Then, we assign labels to it using the inference method of fuzzy logic. Following the completion of the conventional tests by the students, the answer log was used in the RT-based guessing model. The guessing model proposed in this study only uses one parameter, namely time. Further analysis on the answer log data indicated several different behaviours exhibited by the students in giving responses to the presented items. These behaviours occurred due to the duration of the test (90 min), which is long for a multiple-choice test that consists of 40 items. The first behaviour exhibited by the students was that several students used the remaining time to reconsider doubtful responses after they had given responses to all the items. The second behaviour exhibited by the students was that several students spent a lot of time reading items that they deemed difficult, then they skipped the item without giving a response. After giving responses to the other items, the students then came back to the items they deemed difficult and gave a quick response.

Due to these exhibited behaviours, we investigated the use of several parameters to define RT in the proposed guessing model. The first parameter that we used to define RT was the time spent by the students to initially read an item and give a response, which we named duration. The second parameter was the accumulation of time spent on an item even after giving a response, which we named total move. The last parameter, named max time, was derived from the duration parameter, which was the longest time spent to initially read an item and give a response among all the students. We compared the performance of the guessing model with the use of these different parameters.

Table 3 shows the evaluation matrix of threshold determination for the SPM course and SE course. At a glance, the accuracy value of SE course is higher than that of SPM course. This difference is because the number of students taking the exam is not the same. There are more students in the SPM course compared to students in the SE course. This certainly affects the accuracy of the model. The more samples, the greater the potential for outlier behaviour. Therefore, outlier detection [20] is necessary to reduce bias.

**Table 3.** Evaluation of Threshold Determination Methods.

| Couse | Parameter | Method | Accuracy | Precision | Recall | F1 |
|---|---|---|---|---|---|---|
| Software Project Management (SPM) | avg_duration | Common k-second | 66.57% | 16.67% | 5.27% | 8.34% |
| | | Surface | 65.32% | 16.67% | 9.60% | 13.78% |
| | | Normative | 67.01% | 18.85% | 4.33% | 7.04% |
| | | Modified k-second | 68.36% | 16.88% | 2.44% | 4.27% |
| | total_move | Common k-second | 71.14% | - | 0.00% | - |
| | | Surface | 71.03% | 0.00% | 0.00% | - |
| | | Normative | 71.14% | - | 0.00% | - |
| | | Modified k-second | 71.14% | - | 0.00% | - |
| | max_duration | Common k-second | 69.62% | 26.67% | 03.01% | 05.41% |
| | | Surface | 68.58% | 28.44% | 05.84% | 09.69% |
| | | Normative | 70.11% | 28.89% | 02.45% | 04.51% |
| | | Modified k-second | 70.87% | 39.13% | 01.69% | 03.25% |
| Software Engineering (SE) | avg_duration | Common k-second | 84.72% | 16.67% | 1.96% | 3.51% |
| | | Surface | 84.72% | 16.67% | 1.96% | 3.51% |
| | | Normative | 83.33% | 17.86% | 4.90% | 7.69% |
| | | Modified k-second | 85.28% | 16.67% | 0.98% | 1.85% |
| | total_move | Common k-second | 85.83% | - | 0.00% | - |
| | | Surface | 85.83% | - | 0.00% | - |
| | | Normative | 85.83% | - | 0.00% | - |
| | | Modified k-second | 85.83% | - | 0.00% | - |
| | max_duration | Common k-second | 85.83% | 50.00% | 0.98% | 1.92% |
| | | Surface | 85.83% | 50.00% | 0.98% | 1.92% |
| | | Normative | 85.13% | 27.27% | 2.94% | 5.31% |
| | | Modified k-second | 85.83% | - | 0.00% | - |

Each table displays the evaluation matrix of our proposed methods compared to other threshold determination methods. In addition, each table is compared with various aggregation parameters. In general, the guessing model that used the modified k-seconds method to determine the RT threshold outperformed the other models in terms of accuracy. In the SPM course, using the avg_duration, the accuracy was 68% aggregation parameter,

outperforming the other methods. Meanwhile, on the SE course, the accuracy was 85%, outperforming the other methods. Further analysis of modified k-seconds method revealed that the model performed better with the use of the total move and max time parameters. With the use of the total move parameter, the model achieved a higher accuracy. However, this model obtained a recall value of 0. This indicates that the model was unable to detect rapid-guessing behaviour. As a result of the recall metric having a value of 0, the precision and F1 score values were not able to be calculated.

Furthermore, the evaluation of the models based on the F1 score metric revealed that the guessing model that used the surface features method along with the guessing model that used the normative method to determine the RT threshold achieved the best performance. Further analysis of these two models revealed that the performance of both models was more stable with the use of the duration parameter.

Our experiments show that our proposed method, modified k-second, has superior accuracy compared to other methods in both courses. In addition, this study also proves that there is a difference in accuracy along with the difference in aggregation techniques. Aggregation using total_move has higher accuracy than using avg_duration or max_duration parameters. Therefore, further research needs to try other aggregation parameters, one of which is sum_duration. However, when viewed from the F1 score evaluation, the best method is the surface feature. Although in terms of accuracy, modified k-second recorded the highest value, this method has a very low recall value, because the count of students who guessed is very little (data imbalance). This causes the model to be biased, so that the model cannot properly accommodate class with little data [21]. For further research, several techniques need to be conducted to handle data imbalance, such as modifying preprocessing techniques, algorithmic approaches, cost sensitivity, and ensemble learning [21].

## 5. Conclusions

Assessment is used to measure students' abilities with the aim of selecting students for new admissions, measuring the level of understanding of post-learning material, and as a determinant of graduation. However, the results of the assessment may be invalid because the students cheated or rapidly guessed the answer to the question. Rapid-guessing behaviour is a phenomenon where students answer items rapidly without serious thought. Several researchers have conducted studies on how to detect rapid-guessing behaviour by analysing processing time with a certain threshold. However, existing methods have no developed and corrected time threshold determination utilising IRT features and considering variations in data aggregation. Therefore, this study aims to combine the k-second method with IRT features to recognize the difficulty level of each question and utilise multiple data aggregation methods to distinguish rapid-guessing behaviour and solution behaviour. We compared the proposed method and the data aggregation technique. Some of the aggregation techniques we used include average, sum, and maximum. Then, the model is evaluated using accuracy, precision, recall, and F1 score parameters.

This study proves that the correction of threshold determination that we proposed, modified k-second, succeeded in detecting guessing with an accuracy better than the other methods. In SPM courses, modified k-second has an accuracy of 68.36%, superior to other methods using the avg_duration parameter. This research also proves that the selection of aggregation techniques also greatly affects the level of accuracy. Total move is an aggregation parameter that has high accuracy. Meanwhile, average duration is an aggregation parameter that has lower accuracy. However, when viewed from the F1 score evaluation, the best method is the surface feature. Although in terms of accuracy, modified k-second recorded the highest value, this method has a very low recall value, because the count of students who guessed is very little (data imbalance). This causes the model to be biased, so that the model cannot properly accommodate class with little data. For further research, several techniques need to be conducted to handle data imbalance, such as modifying preprocessing techniques, algorithmic approaches, cost sensitivity, and ensemble learning.

**Author Contributions:** Conceptualization, U.L.Y. and A.N.P.B.; methodology, U.L.Y. and A.N.P.B.; software, A.N.P.B.; validation, A.N.P.B. and M.A.; formal analysis, M.A. and E.P.; investigation, A.N.P.B.; resources, A.N.P.B.; data curation, M.A. and E.P.; writing—original draft preparation, M.A.; writing—review and editing, E.P.; visualization, M.A.; supervision, U.L.Y. and E.P.; project administration, U.L.Y.; funding acquisition, U.L.Y. All authors have read and agreed to the published version of the manuscript.

**Funding:** This research was funded by Institut Teknologi Sepuluh Nopember (ITS) for WCP-Like Grant Batch 2, grant number 1855/IT2/T/HK.00.01/2022.

**Data Availability Statement:** The data that support the findings of this study are available from the corresponding author, M.A., upon reasonable request.

**Acknowledgments:** This work is part of the "i-assessment project", an adaptive testing-based test application.

**Conflicts of Interest:** The author declares no conflict of interest.

## Abbreviations

Notation and Acronym

| | |
|---|---|
| RA | Response accuracy |
| RT | Response time |
| RTE | Response time effort |
| IRT | Item response theory |
| SE | Software engineering |
| SPM | Software project management |
| $b_i$ | Item difficulty |
| $N_i$ | The number of examinees that submitted a response item i |
| $n_{fi}$ | The number of examinees that were unable to submit (false answer) a response to item i |
| $l$ | Student's ability |
| $i$ | Rasch model |

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
