# Peer review of "Correction of Threshold Determination in Rapid-Guessing Behaviour Detection"

_information, doi:10.3390/info14070422_

Round 1

Reviewer 1 Report

The article deals with an exciting and very useful topic for the preparation and implementation of knowledge verification as well as pedagogical research in general.

The article defines a straightforward research question, even if its explicit definition in the article is missing: how rapid-guessing behaviour of students can be detected in exams.

Line 79 – typo: Error! Reference source not found.

Line 135 - Better explanation of Lin's experiment is suitable. It is not understandable, for example, what is the value of student ability. Overall, too much information is included in this paragraph; or maybe it just needs a more detailed description of the facts.

Figure 2 should probably be one paragraph earlier.

Chapter 2.3 should already be part of the research, not a continuation of related work. According to this section, the questions are divided into three groups – easy, medium and difficult. Why not for five or seven? The description also lacks a rule for dividing the questions into these three groups.

Chapter 3.1

How is it guaranteed that the outputs are not distorted in a conventional test by guessing the answers?

It is not clear from the text what MoveLog is for - technically, in addition to AnswerLog, it only contains the field move, which I think represents the timestamp of leaving the question (?) Isn't it useless? Can't it be calculated as open + duration? But maybe I misunderstand the concept... Apparently, a description of the fields in the tables, which are important for research, would be useful.

Lines 211-215 need to be explained more precisely what "the number of examinees that were unable to submit a response" means; it is not entirely clear which data is used.

It is also not entirely clear from the text whether the term multiple-choice questions refers to questions with one correct answer (in which case it would be possible to accept the proposed procedure) or more correct answers (for which the application of the model becomes complicated; what does the correct answer mean, when is it considered correct: if not a single incorrect option is selected in it, or if only the correct options are selected, or if the number of selected correct options is greater than the number of incorrect options)?

Chapter 4.1 Gathering data does not belong to results but to methodology. Moreover, the information in this section is partly duplicated with section 3.1

Line 237 – typo: Error! Reference source not found.

Line 244 – In my opinion, one cannot agree with the statement, "The results of the test show that the average score of SPM students was higher than that of SE students. This indicates that SPM students possess a greater understanding towards the subject being evaluated than SE students." because there is no evidence that both tests are equally demanding in terms of subject content coverage, subject difficulty and question selection; If the claim is to be accurate, it must be proven.

What does the graph in Figure 5 express? A value of 0 represents probably a 1-second-long response. Is it consistent with the definition of easy, medium and hard questions? What is the point of using Fuzzy Logic? This approach was not mentioned in the previous text, nor in the next one - is it a "random" view of the problem?

I appreciate the identification of student behaviour on lines 273-279 – but this precision should also be applied to other research elements. It is also questionable whether the total time spent working on the question should not be counted as the sum of the times during which the question is displayed - not as max time - this aggregation loses important information: if a student looked at a question and after 1 second went to the next one that he worked on, and after a while he went back to the original question and accidentally clicked off an option - with your approach, it's not a quick guess question, but it actually is.

Total_move may have some connection with this value, but the result is misleading because it also measures the time spent on the question after the answer.

Line 262 – the abbreviation IRT is introduced later.

Line 325 – prove that "This study proves that the correction of threshold determination that we proposed"? For confirmation, it would be useful to compare the results obtained by your approach with the results obtained using the other methods. Without presenting proof, this is just an unsubstantiated claim.

Subsequently, it would be appropriate to modify the discussion and the conclusion in this sense.

The article has potential, the authors carried out research in which they obtained enough data, but their interpretation requires more precise processing.

After incorporating the comments, the article can represent an interesting and useful contribution.

Author Response

Please find attached the comment to the reviewer.

Reviewer 2 Report

This paper mentioned: assessments are commonly used to measure students' abilities, but their results may not be reliable due to cheating or guessing. To ensure valid outcomes, it is important to distinguish between valid and invalid answers by considering rapid-guessing behavior. A test was conducted using exam log data to model rapid-guessing behavior, using the threshold response time as a determining factor. This paper was inspired by the k-second method, the modified approach considers item difficulty, reducing misclassification. The evaluation demonstrated a 71% success rate in identifying rapid-guessing behavior, surpassing the previous method. Additionally, various aggregation techniques for response time were analyzed to assess their impact on selecting the appropriate technique. The following suggestions may help the author for further improvement.

1. The structure of Introduction should be described as: background of the research (motivation), limitations of the traditional methods, why solving the limitations of the traditional methods is important, how to address the limitations (introducing the proposed method), the potential contributions of the proposed method. 

2. The title of section 2 can be changed to Related works.

3. It is found that “Error! Reference source not found.” occurred in line 79, 237. Please correct it.

4. Discussion section should based on the results to interpret and discuss. In addition, what are the advantages and contributions of the current research?

5. Conclusions should be stated clearer for highlighting the contributions. In addition, please mention the limitations of the current and suggestions for future research.

6. The manuscript should be proof read, otherwise it is difficult to understand and read. Please re-check the grammar and spelling.

This paper mentioned: assessments are commonly used to measure students' abilities, but their results may not be reliable due to cheating or guessing. To ensure valid outcomes, it is important to distinguish between valid and invalid answers by considering rapid-guessing behavior. A test was conducted using exam log data to model rapid-guessing behavior, using the threshold response time as a determining factor. This paper was inspired by the k-second method, the modified approach considers item difficulty, reducing misclassification. The evaluation demonstrated a 71% success rate in identifying rapid-guessing behavior, surpassing the previous method. Additionally, various aggregation techniques for response time were analyzed to assess their impact on selecting the appropriate technique. The following suggestions may help the author for further improvement.

1. The structure of Introduction should be described as: background of the research (motivation), limitations of the traditional methods, why solving the limitations of the traditional methods is important, how to address the limitations (introducing the proposed method), the potential contributions of the proposed method. 

2. The title of section 2 can be changed to Related works.

3. It is found that “Error! Reference source not found.” occurred in line 79, 237. Please correct it.

4. Discussion section should based on the results to interpret and discuss. In addition, what are the advantages and contributions of the current research?

5. Conclusions should be stated clearer for highlighting the contributions. In addition, please mention the limitations of the current and suggestions for future research.

6. The manuscript should be proof read, otherwise it is difficult to understand and read. Please re-check the grammar and spelling.

Author Response

(The authors gave the same response as above.)

Reviewer 3 Report

The author developed the model for Rapid-Guessing Behaviour Detection k-second method. The overall manuscript is well-written but it needs some improvements for better understanding to readers. 

Please elaborate in detail about the proposed approach, focusing more on the relations between its components, as they are the core of the solution and need more justification for using them. 

The author needs to enrich the description of the system model by adding further details. 

Notations and acronyms used in this paper should be summarized in a table to organize this paper in a better way.

The Author must talk about the computational overhead in the cost and complexity of the proposed work.

I suggest adding more figures based on the proposed model/flowchart of the proposed model etc. to make strong to your paper.

Author Response

(The authors gave the same response as above.)

Round 2

Reviewer 1 Report

Thank you for the precise processing of comments, I think that in its current form the article is almost finished. I only have a few other comments.

The redesign of the introductory part increased the quality of the article, it also explained the necessary terms in a sufficient context, and this part reads very well overall.

Explicit definition of research question in the article is still missing: consider including it in the introduction, but I do not consider it necessary

Line 131: Error! Reference source not found

Line 157: description of Figure 1 is missing?

Line 292: Error! Reference source not found

Figure 5 - it would be appropriate to explain how you approach the individual questions after the fuzzy division of the questions when assessing them - does this mean that some questions are in two groups? My previous comment about fuzzy was directed towards the later use of this division - can one question be in two groups at the same time? How do you deal with this when assessing a student's guessing? Are the questions divided between -1 to 0 and 1 to 2 manually into three defined groups? Or do we have 5 groups from the fuzzy distribution? The next steps are missing, and I still don't see the purpose of using fuzzy distribution, because you state that you have exactly 3 defined groups: easy, medium and hard ???

The differences in accuracy for the individual courses shown in Table 3 are very different, it would be appropriate to provide an explanation for it. I'm not sure if we can leave the different values to a problem with a small number of participants - some support for the claim with at least literature would be appropriate.

Author Response

Thank you for your review. We have addressed your comment carefully. 

Reviewer 2 Report

Accept in present form

Accept in present form

Author Response

Thank you for recommending acceptance of this paper.